# A Retrospective Medical Record Review to Describe Health Status and Cardiovascular Disease Risk Factors of Bus Drivers in South Africa

**DOI:** 10.3390/ijerph192315890

**Published:** 2022-11-29

**Authors:** Susan C. Aitken, Samanta T. Lalla-Edward, Maren Kummerow, Stan Tenzer, Bernice N. Harris, W. D. Francois Venter, Alinda G. Vos

**Affiliations:** 1Genesis Analytics, Johannesburg 2196, South Africa; 2School of Health Systems and Public Health, University of Pretoria, Pretoria 0002, South Africa; 3Ezintsha, Faculty of Health Sciences, University of the Witwatersrand, Johannesburg 2000, South Africa; 4Julius Global Health, Julius Center for Health Sciences and Primary Care, University Medical Center Utrecht, Utrecht University, 3584 CX Utrecht, The Netherlands; 5Farraday Medical Centre, Johannesburg 2001, South Africa

**Keywords:** cardiovascular disease, obesity, hypertension, diabetes, road transport

## Abstract

Cardiovascular disease (CVD) is the leading cause of death globally. The occupational challenges of bus drivers may increase their risk of CVD, including developing obesity, hypertension, and diabetes. We evaluated the medical records of 266 bus drivers visiting an occupational medical practice between 2007 and 2017 in Johannesburg, South Africa, to determine the health status of bus drivers and investigate risk factors for CVD, and their impact on the ability to work. The participants were in majority male (99.3%) with a median age of 41.2 years (IQR 35.2); 23.7% were smokers, and 27.1% consumed alcohol. The median body mass index (BMI) was 26.8 m/kg^2^ (IQR 7.1), with 63.1% of participants having above normal BMI. Smoking, BMI, and hypertension findings were in line with national South African data, but diabetes prevalence was far lower. Undiagnosed hypertension was found in 9.4% of participants, uncontrolled hypertension in 5.6%, and diabetes in 3.0%. Analysis by BMI category found that obesity was significantly associated with increased odds of hypertension. Uncontrolled hypertension was the main reason for being deemed ‘unfit to work’ (35.3%). Our research highlights the need for more regular screening for hypertension and interventions to address high BMI.

## 1. Introduction

The World Health Organization (WHO) ranks cardiovascular disease (CVD) as the leading cause of death globally [1]. Commercial drivers, including bus drivers, experience physical and mental challenges associated with the nature of their occupation, extended periods of time in a sedentary position with limited ability to take standing breaks, irregular break times, unhealthy diet, extended periods of time away from home, and limited access to health services whilst travelling [2,3,4,5,6,7]. In addition to physical challenges, commercial drivers have high mental demands, including disrupted sleep patterns, strenuous driving conditions, such as traffic and night driving, loneliness and isolation, crime risks, and passenger communication [8,9,10,11,12,13,14,15,16,17,18,19]. These conditions can lead to the development of risk factors associated with CVD, namely obesity, hypertension, and diabetes [2,3,10,12,13,14,17,20]. Increased obesity rates for long distance drivers, including truck and bus drivers have been reported [21,22,23,24,25]. Additional risk factors that are prevalent in commercial drivers include the use of alcohol and tobacco [4,5,26].

There are currently little to no routine occupational checks in this context, aside from an annual check-up for a professional drivers permit (PDP), which requires bus drivers to know their health profile. Irregular work hours and an inability to predict attendance at scheduled appointments may additionally impact clinic attendance [3,19,27]. Risk factors, such as abnormal glucose profiles and hypertension, are often only diagnosed during research [28]. These risk factors remain undiagnosed, and this highlights the need for improved availability of health care services for bus drivers.

To develop health interventions that best serve the needs of bus drivers in South Africa, research investigating the status of health and the presence of CVD risk factors is needed to better understand their specific health concerns.

Improving bus driver health is vital for their own safety and well-being, as well as those of passengers and other road users. There is currently no published health status data about this group for South Africa. Understanding the health issues that bus drivers face is needed to better develop health interventions, including modifying working conditions to support health needs.

In this paper, we describe the health status of bus drivers in South Africa based on data compiled from a retrospective medical record review. The main objective is to determine the health status of bus drivers with a special focus on risk factors for CVD. Furthermore, we investigated whether any of the health conditions affected their ability to work. Lastly, we looked for associations between CVD risk factors.

## 2. Materials and Methods

### 2.1. Study Design and Setting

In South Africa, employers are required to adhere to the regulations of the Occupational Health and Safety Act 85 of 1993, which states that employers who employ people to conduct certain listed work must conduct periodic examination of these employees. These examinations include clinical or medical tests, which must be conducted by an occupational medical practitioner or a person who holds a qualification in occupational health, recognized as such by the South African Medical and Dental Council [29]. The purpose of the act is to ensure the health and safety of employees and persons connected to the activities of persons at work. As such, all commercial bus drivers in South Africa are required to have regular formal medical assessments to operate vehicles.

This study was a secondary data analysis conducted between 2018 and 2021 on occupational health assessment data collected for Greyhound bus drivers at the Faraday Medical Centre in Johannesburg, South Africa. The Faraday Medical Centre is concerned with the occupational evaluation of employees in the road and aviation sectors. They have a long-standing history of providing driver medicals for a number of large transport companies, including long haul drivers and passenger bus drivers. All assessments conducted are performed by a medical doctor with a diploma in occupational health, as well as a certification in medical evaluation of professional drivers.

Greyhound is one of the largest bus companies providing passenger transport services between major cities in South Africa. Greyhound and subsidiary companies are part of the Unitrans Passenger, owned by KAP Industrial Holdings Limited. Based on the KAP 2018 integrated report, Unitrans Passenger services had 4063 employees, 1419 revenue earning vehicles, travelled more a 108 million kilometres per annum, and transported more than 1.3 million intercity passengers per annum.

Due to the long-standing relationship between Greyhound and the Faraday Medical Centre, a substantial number of medical records were available spanning many years, and these data were collected consistently by the same medical professional. For this reason, this research focuses on bus drivers who were employed by Greyhound and their subsidiary companies including: Mega Coach, Mega Bus, Magic Transfers, Unitrans, Mega Express, and Citiliner.

### 2.2. Inclusion/Exclusion Criteria

Records were included if the individual was a bus driver aged 18 years and above, with at least two or more visits that were 12 months apart, and for whom all critical data parameters were available, which included age, gender, and occupation. All eligible records in the period from 2007 to 2017 were included.

### 2.3. Measurements

Data were extracted from the occupational medical reports and captured electronically using Research Electronic Data Capture (REDCap, Vanderbilt University, USA). During data collection, each individual was assigned an anonymous participant identifier. Demographic data was limited to sex and date of birth. Data on ethnicity, marital status, income, and education was not available as it was not required for the occupational health assessment. Mental health assessments were also not conducted.

Alcohol use, smoking, and medical history were self-reported. Physical examination included measuring height, weight, heart rate, blood pressure, as well as an assessment of heart, lungs, abdomen, gait, central nervous system (CNS), and balance. Basic biomedical testing included blood glucose and urinalysis. Functional testing comprised of stress electrocardiogram (ECG), exercise tolerance, and lung function tests, including peak expiratory flow (PEF), and oxygen saturation (Sp0_2_).

Current medical conditions were determined based on measurements during the consultation, as well as notes of newly diagnosed conditions made by the occupational health doctor.

BMI was calculated using the standard formula (weight [kilograms, kg]/height squared [meters, m, squared]) and categorized into four standard groups: underweight BMI, <18.5 kg/m^2^; normal, 18.5 kg/m^2^ to <25 kg/m^2^; overweight, 25 kg/m^2^ to <30 kg/m^2^; obese ≥30 kg/m^2^.

Blood pressure was measured using both manual and automated methods. For the automated method, a Welch Allen automated blood pressure machine was used. Manual measurements were taken in instances where automated was not possible, such as no power for the automated machine. A normal blood pressure was a systolic blood pressure (SBP) of ≤139 mmHg and a diastolic blood pressure was (DBP) ≤89 mmHg, and hypertension was defined as a SBP of ≥140 mmHg and/or DBP ≥ 90 mmHg [30]. Individuals were classified as having hypertension based on self-reported medical history, if a follow-up visit still met the criteria, or if the doctor noted a diagnosis of hypertension at any moment.

A rapid blood glucose test was performed as part of the occupational health assessment. The WHO diagnostic criteria for diabetes mellitus (DM) was used, defined as a fasting plasma glucose ≥7.0 mmol/L (126 mg/dL) or random glucose ≥11.1 mmol/L (200 mg/dL) [31]. Individuals were considered to have DM based on self-reported medical history, or a new diagnosis noted by the doctor during the occupational health assessment based on the glucose.

All ECG testing and reviewing of results was conducted by the occupational health doctor. If any issues were identified during testing, the ECG would be sent to a cardiologist for further assessment. 

All individuals were observed whilst walking on a treadmill at a consistent pace, and exercise tolerance was defined as good, moderate, or poor, based on the clinical judgement of the occupational health doctor. The clinical picture of each bus driver assessed was considered to make recommendations to the employer. If there were no notable concerns, a driver was deemed ‘fit to work without restriction.’ If one or more concerns were noted, a driver was deemed ‘fit to work with restrictions,’ which included a referral for further treatment or testing and follow-up a few days later, or more regular screening for certain conditions. If serious concerns were noted, a driver was deemed ‘unfit to work’ and removed from duty until the concerns were addressed and the driver was re-assessed.

### 2.4. Statistical Analysis

Data was exported from REDCap and analyzed using Stata 15.1^®^ (StataCorp LLC, College Station, TX, USA).

For descriptive statistics, continuous variables were summarized by medians with interquartile range if skewed and means with standard deviations if normally distributed. Categorical variables were summarized by frequency counts and percentages. The 95% confidence intervals were calculated for incidence rates as applicable.

As BMI is known to be a main CVD risk factor and has known associations with most other CVD risk factors, in a further analysis the association between CVD risk factors was analyzed against BMI in three categories (normal, overweight, and obese), with normal BMI used as the reference for analysis.

In a second analysis, demographic and health data for overweight and obese participants were compared to normal BMI. Continuous data were compared with the Student’s *t*-test or Wilcoxon rank sum test, depending on the Normality assumption. Categorical variables were tested with the Chi-square test or Fisher exact test, as appropriate.

Finally, univariate logistic regression models were used to test for the association between four key CVD risk factors, namely, smoking, alcohol use, hypertension, and diabetes, and BMI category (normal (reference), overweight, and obese) as outcome. All results were reported with their respective 95% confidence intervals. All statistical tests were considered significant where the *p*-value was less than 0.05.

## 3. Results

### 3.1. Description of Study Population

Records of 266 bus drivers were included. Baseline visits for participants meeting the inclusion criteria (*n* = 266) were conducted between 2 February 2007 and 15 November 2017. Demographics, medical history, health status, and CVD risk factors are summarised in Table 1.

The study population was in majority male (99.3%; 264/266) with a median age of 41.2 years (IQR 35.2). The mean height of participants was 1.7 m (SD 0.1); the median weight of participants was 80.0 kg (IQR 77.0). A total of 23.7% (63/266) reported that they were smokers, and 27.1% (72/266) reported they consumed alcohol, all of whom reported only consuming it socially.

### 3.2. Health Status of Study Participants

In the self-reported medical history, hypertension was mentioned most often (10.5%), while none reported a diagnosis of dyslipidaemia. For participants who reported a history of tuberculosis, 57.1% were currently on treatment (4/7), and for 2 participants tuberculosis was suspected at baseline and confirmed with treatment at the next visit. A CVD-related diagnosis was reported for 7 participants, including ischemic changes in previous ECG (*n* = 2), cardiac failure (*n* = 2), stroke, myocardial infarction, and coronary artery bypass graft surgery (all *n* = 1).

More than half of the participants were overweight or obese (168/266, 63%). Hypertension was present in 19.5% of participants. Of these, 53.8% (28/52) had reported a medical history of hypertension, whilst the remaining 46.1% (25/52) were undiagnosed. Of those with a history of hypertension, 53.6% (15/28) had uncontrolled hypertension at the time of assessment. DM was reported or found in 8 participants (3.0%).

Abnormalities that were noted during stress ECG included ischemic changes (5/22), left ventricular hypertrophy (LVH; 4/22), both ischemic changes and LVH (1/22), and right bundle branch block (1/22). During exercise tolerance testing, dyspnoea and/or chest pain was noted for 1.9% (5/265) of the participants.

The vast majority of participants were deemed ‘fit to work without restrictions’ (87.2%), while 6.4% were deemed ‘unfit to work’ (17/266). The primary cause for a participant to be ‘unfit to work’ was uncontrolled hypertension (35.3%, 6/17), followed by uncontrolled blood glucose (17.6%, 3/17), and TB (17.6%, 3/17).

### 3.3. Comparison of CVD Risk Factors by BMI Category

Of the 266 participants, 262 were evaluated in their respective BMI category for associations for CVD risk factors and abnormal findings on testing; those with underweight BMI (4/266; 1.5%) were excluded from the analysis. All variables that were compared are summarized in Table 2.

There were significantly fewer smokers amongst the obese participants (14.6%; 12/82; *p* = 0.02).

Hypertension was significantly more prevalent in overweight (22.1%; 19/86; *p* = 0.005) and obese participants (31.7%; 26/82; *p* < 0.001), than in participants with a normal BMI (7/94; 7.4%).

There were more abnormal ECG assessments for overweight (12.8%; 11/86) and obese participants (11.0%; 9/82) compared to participants with a normal BMI (3.2%; 3/94); this difference was only significant for overweight participants (*p* = 0.02). Good exercise tolerance was highest for participants with a normal BMI (95.7%; 90/94), followed by overweight participants (93.0%; 80/86), but significantly lower for obese participants (86.6%; 71/82; *p* = 0.03).

The effect of health on the ability to work was similar across categories. The only significant difference was for the category ‘fit to work with restrictions’, in which more obese participants were restricted (12.2%; 10/82; *p* = 0.008) than those with a normal BMI (2.1%; 2/94).

### 3.4. Association of CVD Risk Factors by BMI Category

The association of the four key risk factors, smoking, alcohol use, hypertension, and diabetes are given in Table 3**.** Logistic regression indicated that the odds of smoking in obese participants was less than half that of participants who had a normal BMI, producing a statistically significant difference. Participants who were overweight were 3.5 times more likely to have hypertension (OR = 3.5; CI = 1.4–8.9; *p* = 0.007), and those who were obese were 5.8 times more likely to have hypertension (OR = 5.8; CI = 2.3–14.2; *p* < 0.001), than participants with a normal BMI. There were no significant associations between BMI categories and alcohol use or diabetes.

## 4. Discussion

This study investigated the health status and CVD risk factors in bus drivers in South Africa, as well as the impact of these factors on their ability to work. The findings indicate that in general the health status is good, although there is a high prevalence of overweight and obesity, with 63.1% of participants having an above normal BMI. An increased BMI is associated with increased odds of hypertension, another major risk factor for CVD. Additionally, uncontrolled hypertension and diabetes were reasons for drivers to be deemed ‘unfit to work’.

Our BMI data aligns with previous data on long-distance drivers, where between 56.8% and 79.2% were reported to be either overweight or obese [21,22,23,24,25]. In addition to excess weight being linked to risk factors for CVD, Mansur et al. (2015) have also suggested that obesity is related to poor sleep quality and increased sleepiness [26], placing drivers at risk of having accidents. Increased risk of accidents correlates to drivers with stress problems, and that stress was associated with high blood pressure [5].

The frequency of hypertension was relatively low (19.6%) compared to other bus driver populations, such as Nigeria (39.7%; N = 293) [28], Ghana (38.7%; N = 527) [32], Brazil (45.2%; N = 250) [22], and Korea (53.3%; N = 443) [33]. This was also low compared to hypertension data from SANHANES (N = 16,293), which found that the prevalence of hypertension was 35.1% among South Africans aged 15 years and older [34], and the WHO Strategic Advisory Group of Experts (SAGE) South Africa Wave 2 study (N = 2761), which found prevalence to be 23.2% [35]. A possible explanation for this is the healthy worker selection effect, as candidate drivers being screened for fitness to work may have been deemed ‘unfit to work’ due to uncontrolled hypertension. These individuals would therefore not have had a second visit and as such would not have been included in the analysis. This is supported by the finding that the primary reason for being deemed ‘unfit to work’ was hypertension.

Although the prevalence of hypertension was low, the association with increased BMI was still significant. This association has been shown in many other studies. Amadi et al. (2018) reported a nearly three times increased odds of hypertension in those who were overweight/obese. Data from SANHANES also found that hypertension was highest for those who were overweight (48.3%; N = 16,293) [36]. A recent study on CVD risk factors in long-distance bus drivers in Lagos, Nigeria (N = 293), reported that BMI and an extended length of time as a professional driver correlated with the risk of hypertension [28]. These findings were similar for truck drivers in Brazil, Egypt, and America [5,28,37].

Importantly, in our study, almost half of those with hypertension were unaware of their condition at the time of their assessment, and 53.6% of those that were previously diagnosed with hypertension had uncontrolled hypertension at the time of assessment. Similar findings were reported in SANHANES [34]. This highlights the need for improved occupational screening in all sectors, not only those with high-risk occupations.

Compared to the prevalence of diabetes found in other studies focussing on bus drivers, including Nigeria (13.0%; N = 293) [28] and Korea (26.8%; N = 443) [33], the prevalence of diabetes in our study was low (3.0%), but similar to data from Brazil (2.8%; N = 659) [38]. Whilst no explanation was given in the Brazil study, ours showed that uncontrolled diabetes was a reason for drivers to be restricted from work and, as with hypertension, may have been screened out of employment due to the healthy worker selection effect.

The findings on alcohol use and smoking in this investigation align with the population level data for alcohol consumption (27.1% versus 24.2%), and the cohort-level data for smoking (23.7% versus 22.0%) [39,40]. There is ample data that smoking is associated with lower BMI, including findings from our region, as well as findings in truck drivers [21,41,42,43]. Our findings were in line with this, in that significantly fewer individuals with obesity smoked compared to those with a normal BMI.

### Strengths and Limitations

A major strength of this analysis is the consistency with which data was collected over time at a single facility and by the same medical professional. Systematic assessments allowed us to include participants with follow-up visits to provide detailed insights on health status over time and the impact on work. Another strength is the presentation of real-world data from drivers in one of the largest bus companies in South Africa, making our results generalizable to other drivers with the same working conditions in the country.

Limitations include that data was retrospective. Data compared was part of occupational health screening required by the employer, with probable healthy worker selection bias. We additionally did not have data on length of time as a bus driver, social determinants of health, workload, psychological data, or any interventions or work health programmes that drivers might have been part of.

## 5. Conclusions

Overall, bus drivers were in relatively good health, although our data shows high levels of overweight and obesity, as well as high levels of both undiagnosed and under-treated hypertension, and high smoking levels, but this was all similar to the general population. We found surprisingly low levels of diabetes, reported cardiovascular disease, or objective evidence of ischaemic heart disease on ECG or stress testing. Uncontrolled hypertension and DM influenced the ability of bus drivers to work. These key findings need to be taken into account when developing health interventions to improve health and employability of working people with sedentary jobs, including bus drivers.

Prospective research that focusses on diet, exercise, and mental health of bus drivers could provide additional information as to the high prevalence of above normal BMI, and further inform how this could be integrated into workplace wellness programmes.

Occupational health screening programmes provide vital research opportunities for monitoring non-communicable and other conditions, as well as potential referral pathways. In countries such as South Africa, where significant progress has been made in the prevention and treatment of HIV and TB, attention has shifted to the major burden of obesity, diabetes, and hypertension, and associated conditions such as sleep apnoea. Screening, referral, and treatment is vital to further improve health outcomes in these working populations.

## Figures and Tables

**Table 1 ijerph-19-15890-t001:** Baseline demographics and occupational health assessment findings of bus drivers in South Africa, 2007 to 2017.

**Demographics**	
Male, *n* (%)	264 (99.3)
Age (years), median (IQR)	41.2 (35.2)
Height (m), mean (SD)	1.7 (0.1)
Weight (kg), median (IQR)	80.0 (77.0)
**Behavioural Risk Factors**	
Smoker, *n* (%)	63 (23.7)
Alcohol use ^1^, *n* (%)	72 (27.1)
**Reported Medical History**	
Hypertension, *n* (%)	28 (10.5)
Diabetes, *n* (%)	7 (2.6)
Tuberculosis, *n* (%)	7 (2.6)
CVD-related ^2^, *n* (%)	7 (2.6)
Physical injury, *n* (%)	4 (1.5)
Dyslipidaemia, *n* (%)	0 (0.0)
Other ^3^, *n* (%)	2 (0.8)
**BMI**	
BMI (m/kg^2^), median (IQR)	26.8 (7.8)
**BMI Category, *n* (%)**	
Underweight	4 (1.5)
Normal weight	94 (35.4)
Overweight	86 (32.3)
Obese	82 (30.8)
**Hypertension**	
SBP (mmHg), median (IQR)	123.5 (82.0)
DBP (mmHg), median (IQR)	80.0 (40.0)
Hypertension, *n* (%)	52.0 (19.5)
**Diabetes**	
Diabetes, *n* (%)	8 (3.0)
Blood glucose (mmol/litre), median (IQR)	5.2 (8.8)
**General Health, *n* (%)**	
Heart rate (bpm), median (IQR)	66.0 (37.0)
Urine dipstick (normal)	251 (94.4)
Heart & lungs (normal)	259 (97.4)
CNS (normal)	265 (99.6)
Gait (normal)	264 (99.3)
**Stress ECG Measurements, *n* (%)**	
Outcome (Normal)	243 (91.4)
**Exercise tolerance**	
Good	245 (92.1)
Moderate	12 (4.5)
Poor	9 (3.4)
Dyspnoea (Without)	260 (98.1; N = 265)
**Outcome of Assessment, *n* (%)**	
Fit to work without restrictions	232 (87.2)
Fit to work with restrictions	17 (6.4)
Unfit to work	17 (6.4)

^1^ Social alcohol use; ^2^ Includes: ischemic changes in ECG, cardiac failure, stroke, myocardial infarction, cardiac bypass surgery; ^3^ Includes: Bell’s Palsy, and ovarian cysts.

**Table 2 ijerph-19-15890-t002:** Comparison of overweight and obese bus driver demographic and health data versus normal BMI.

	Normal ^1^	Overweight	*p*	Obese	*p*
Total (N = 262), *n* (%)	94 (36.1)	86 (33.7)	-	82(32.2)	-
Male, *n* (%)	94 (100)	85 (98.8)	0.5 *	81 (98.8)	0.5 *
Age, mean (SD)	39.8 (8.4)	43.6 (9.5)	0.003 ^	43.5 (9.2)	0.004 ^
Smoker, *n* (%)	28 (29.8)	21 (24.4)	0.4 *	12 (14.6)	0.02 *
Alcohol use ^1^, *n* (%)	25 (26.6)	21 (24.4)	0.7 *	24 (29.3)	0.7 *
Diabetes, *n* (%)	2 (2.1)	3 (3.5)	0.7 **	3 (3.7)	0.7 **
Blood glucose, median (IQR)	5.0 (4.0)	5.2 (3.8)	0.44 ^^	5.4 (3.3)	0.02 ^^
Hypertension, *n* (%)	7 (7.4)	19 (22.1)	0.005 *	26 (31.7)	<0.001 *
SBP, median (IQR)	120 (54)	130 (46)	<0.001 ^^	130 (60)	<0.001 ^^
DBP, median (IQR)	80 (30)	80 (21)	<0.001 ^^	80 (23)	<0.001 ^^
ECG abnormal, *n* (%)	3 (3.2)	11 (12.8)	0.02 *	9 (11.0)	0.07 *
Exercise tolerance					
Good, *n* (%)	90 (95.7)	80 (93.0)	0.5 **	71 (86.6)	0.03 *
Moderate, *n* (%)	2 (2.1)	3 (3.5)	0.7 **	7 (8.5)	0.1 **
Poor, *n* (%)	2 (2.1)	3 (3.5)	0.7 **	4 (4.9)	0.4 **
Dyspnoea (Without)	92.0 (98.9)	85 (98.8)	1.0 **	79 (96.3)	0.3 **
Fit to work without restrictions, *n* (%)	83 (88.3)	79 (91.9)	0.4 *	67 (81.7)	0.2 *
Fit to work with restrictions, *n* (%)	2 (2.1)	4 (4.6)	0.4 **	10 (12.2)	0.008 *
Unfit to work, *n* (%)	9 (9.5)	3 (3.5)	0.1 *	6 (6.1)	0.4 *

^1^ Reference; ^ *t*-test; ^^ Wilcoxon rank sum test; * Chi-squared; ** Fisher’s Exact.

**Table 3 ijerph-19-15890-t003:** Association of cardiovascular risk factors by BMI category for bus drivers determined using logistic regression analysis.

	OR (95%CI; *p*-Value) *	
	Overweight	Obese
Smoker	0.8 (0.4–1.5; 0.5)	0.4 (0.2–0.9; 0.02)
Alcohol use	0.9 (0.5–1.7; 0.7)	1.1 (0.6–2.2; 0.7)
Hypertension	3.5 (1.4–8.9; 0.007)	5.8 (2.3–14.2; <0.001)
Diabetes	1.7 (0.3–10.2; 0.6)	1.7 (0.3–10.7; 0.5)

* Normal BMI used as the reference. Abbreviations—OR: odds ratio; CI: Confidence interval; *p*: probability.

## Data Availability

The data presented in this study are available on request from the corresponding author.

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
