# Peer review of "A Retrospective Medical Record Review to Describe Health Status and Cardiovascular Disease Risk Factors of Bus Drivers in South Africa"

_ijerph, 2022, doi:10.3390/ijerph192315890_

Round 1

Reviewer 1 Report

Despite being very interesting, the paper lacks a better-described practical implication. Understanding the role of occupation in health is imperative, but the design was retrospective and merely descriptive. The authors presented the data with overall merit, including a well-structured methods section. However, the paper needs some revisions. 

I strongly recommend that the authors carefully check the writing, including minor spelling errors and some long sentences that lack clarity. 

Title

Regarding the title, it could be interesting to add the study design. 

Abstract

The abstract lacks the purpose of this study, as well as its design. The methods were poorly reported in the abstract, which compromised the understanding of the study and its main findings. The authors intended to describe the profile of South African bus drivers or to investigate whether being a bus driver was associated with more cardiovascular risk and worse health status. Based on these aspects, the whole abstract should be improved.

Introduction

The introduction must be linked to the reporting data. In addition, the purpose of this study remained unclear in the introduction, as well as it was in the abstract. These two paragraphs described several occupational aspects that were not investigated. Therefore, the manuscript requires a major revision.

Methods

This section was well-structured and only require minor revisions. I suggest revising the whole section to ensure clarity, mainly regarding some long sentences, for instance, lines 56-60.

It is worth noting that the authors choose to include only data from bus drivers from Greyhound and their subsidiaries, but it is unclear the reason for those criteria. In addition, the authors did not provide occupational data that could help to explain the findings.  

The description of the statistical analysis does not match the purpose of the study. Therefore, it is important to provide a better description of both the objective and the statistical approach to attend to it.

Results 

Similar to the methods section, the results were well-presented. However, occupational and demographic (age, race, ethnicity, gender, marital status, income, education, and employment) data are missing. 

Another aspect that needs revision is the excessive use of ; throughout the section. Despite being well-structured, the results were repetitive, except for 3.3. The authors should provide some prior interpretation of the findings.  

The authors need to cite the Tables in the description of the results. Except for Table 1, the other two tables were not even mentioned. The titles from tables 2 and 3 could also be improved by adding the statistical analysis performed. I strongly recommend separating the comparison data from the association, as well as providing a clearer interpretation of the results available in the tables. 

Discussion

The lack of clarity regarding the purpose of this study made the discussion confusing and very similar to the introduction. The discussion did not help the reader to understand the main findings and also does not explain them. This section needs a major revision. I suggest resuming the purpose of this study followed by its main implications. Then, the authors could briefly report the main findings followed by a discussion based on the same structure of the results section. I also missed a discussion regarding the social determinants of health and the role of inequalities. 

Limitations

The addition of limitations as a section helps to enhance the overall quality of the manuscript. It could be interesting to point out the strengths. Since the authors choose to investigate bus drivers, another potential limitation is the absence of occupational data (i.e., workload), as well as possible coping strategies. 

Conclusion

The conclusion must be improved, especially regarding the writing to provide more clarity. In addition, the conclusion section must be in accordance with the purpose of this study. Therefore, this section requires revision, but I recommend revising it after checking the previous comments. 

Reviewer 2 Report

The ijerph-1986294 manuscript is interesting and aims to relate different physical parameters with the risk of cardiovascular diseases in a specific group of subjects, ie bus drivers.

This particular category of subjects is exposed to different working conditions which increase their sedentary lifestyle and at the same time increase the cardiovascular risk factor.

Despite these premises, however, there is no clear indication in these subjects that their specific job role exposes them to this risk.

However, there are some critical issues that need to be clarified before being able to consider this manuscript for publication.

It is definitely a good thing that this category of workers is quite fit.

Certainly the company's occupational doctor screening has helped "select" the healthiest drivers. This should be emphasized.

It appears that these subjects have an average BMI of 26.8 but analyzing the percentage it appears that overweight subjects (> 25) are 32% and obese subjects (> 30) are 31%.

So overall, the subjects with a BMI greater than 25 are 63%. Then recheck the average BMI.

The subjects considered are long-distance bus drivers, who are subject to a sedentary lifestyle but much less stress than a driver who drives into a city or a metropolis. This fact is increased and widely commented on by virtue of the difference in physical-physical stress suffered.

Double check lines 190-193, there are terms in Chinese.

Round 2

Reviewer 1 Report

I appreciate your responses. However, the paper needs a minor revision for checking the English writing and compliance with journal instructions for authors.

The changes were made in accordance with the requests, but  I strongly recommend that the authors mention the occupational data as a perspective for future research, as well as possible strategies for screening risks and referring to healthcare.

Author Response

We have added a final sentence to the manuscript. “Occupational health screening programmes provide vital research opportunities for monitoring non-communicable and other conditions, as well potential referral pathways. In countries like South Africa, where significant progress has been made in the prevention and treatment of HIV and TB, attention has shifted to the major burden of obesity, diabetes and hypertension, and associated conditions such as sleep apnoea. Screening, referral and treatment is vital to further improve health outcomes in these working populations.”

Reviewer 2 Report

I thank the authors for responding exhaustively to most of the criticisms raised.

It is my opinion that the manuscript should emphasize, perhaps in the Limitation/Conclusions section, that this group of subjects has in fact already been selected for employment by a medical screening.

That is, it is an "elite" group among bus drivers.

A doubt remains for me of the great difference that exists between this group of subjects and the drivers of buses on the metropolitan line.

Is the fact that they are so fit, despite the concrete possibility of an increase in cardiovascular risk factors, due to selection or eating habits or physical activity?

Author Response

We have added an important clarification to the end of the first paragraph of “Study design and setting” – these are not ‘elite’ drivers, all drivers require medical certificates, so we have added “All commercial bus driver in South Africa are required to have regular formal medical assessments to operate vehicles.” We hope this addresses the issue.